**Data Availability Statement:** All relevant data are available at Github: https://github.com/li-lab-mcgill/recurrent-disease-progression-networks.

**Funding:** YL is supported by Natural Sciences and Engineering Research Council (NSERC) Discovery

# Recurrent disease progression networks for modelling risk trajectory of heart failure

Xing Han Lu[1]☯, Aihua Liu[2]☯, Shih-Chieh Fuh[1], Yi Lian[3], Liming Guo[2], Yi Yang[4], Ariane Marelli[2]*, Yue Li[1]*

1 School of Computer Science, McGill University, Montreal, Canada, 2 McGill Adult Unit for Congenital Heart Disease Excellence (MAUDE Unit), Montreal, Canada, 3 Department of Epidemiology, Biostatistics and Occupational Health, McGill University, Montreal, Canada, 4 Department of Mathematics and Statistics, McGill University, Montreal, Canada

☯ These authors contributed equally to this work.
* ariane.marelli@mcgill.ca (AM); yueli@cs.mcgill.ca (YL)

## Abstract

### Motivation

Recurrent neural networks (RNN) are powerful frameworks to model medical time series records. Recent studies showed improved accuracy of predicting future medical events (e.g., readmission, mortality) by leveraging large amount of high-dimensional data. However, very few studies have explored the ability of RNN in predicting long-term trajectories of recurrent events, which is more informative than predicting one single event in directing medical intervention.

### Methods

In this study, we focus on heart failure (HF) which is the leading cause of death among cardiovascular diseases. We present a novel RNN framework named Deep Heart-failure Trajectory Model (DHTM) for modelling the long-term trajectories of recurrent HF. DHTM auto-regressively predicts the future HF onsets of each patient and uses the predicted HF as input to predict the HF event at the next time point. Furthermore, we propose an augmented DHTM named DHTM+C (where "C" stands for co-morbidities), which jointly predicts both the HF and a set of acute co-morbidities diagnoses. To efficiently train the DHTM+C model, we devised a novel RNN architecture to model disease progression implicated in the co-morbidities.

### Results

Our deep learning models confers higher prediction accuracy for both the next-step HF prediction and the HF trajectory prediction compared to the baseline non-neural network models and the baseline RNN model. Compared to DHTM, DHTM+C is able to output higher probability of HF for high-risk patients, even in cases where it is only given less than 2 years of data to predict over 5 years of trajectory. We illustrated multiple non-trivial real patient examples of complex HF trajectories, indicating a promising path for creating highly accurate and scalable longitudinal deep learning models for modeling the chronic disease.

Grant (RGPIN-2019-0621), Fonds de recherche Nature et technologies (FRQNT) New Career (NC-268592). The Québec CHD Database is funded by Dr. Marelli's Canadian Institute of Health Research Foundation Grant Award #35223.

**Competing interests:** The authors have declared that no competing interests exist.

# 1 Introduction

Heart failure (HF) is leading cause of death among patients with congenital heart disease (CHD) [1]. In the past few decades, research on heart disease [2] has significantly advanced along with the advance of biomedical sciences, while long-term prediction of HF remains a challenge due to its complexity in clinical and epidemiological aspects. Since 1990, there have been at least 19 studies that proposed different models in HF predictions. As recently reviewed [3], many of these methods have limited capacity to model the complex HF events, which resulted in poor generalizable performances. The key challenges do not only include the lack of standardized clinical data on a large patient cohort but also the lack of a systematic and generalizable approach that takes all patient information and makes unbiased predictions of the future risk of HF for any given patient.

With the advance in the field of electronic health record (EHR), HF prediction models can be built on a large amount of administrative data. Data from clinical tests can be complemented with administrative data [4]. In addition, the relationship among different co-morbidities (CM) is difficult to capture by merely mining the large amount of EHR data, and it is only recently that deep learning models have gained the attention for prediction tasks in medicine [5–7].

There is a tremendous opportunity to improve CHD-related HF by modeling long-term patient medical history. In addition to genetic components, CM and surgeries play important roles in the pathogenesis of HF. However, there is a lack of efficient method that can leverage the long-term patient history to make accurate prediction of HF. To address this challenge, we propose a deep Recurrent neural network (RNN) model that models not only the HF trajectory but also the longitudinal data from EHR on the acute co-morbidity diagnoses. Our model is inspired by the sentence completion models in natural language processing (NLP) [8]. In the NLP application, the RNN is provided with the first few words to complete the sentence by predicting subsequent words one at a time. When predicting the $(t + 1)^{th}$ word, the model uses as input the *predicted $t^{th}$* word together with the input gate activities. Analogously, to predict long term-trajectory of HF, we use the predicted HF at future time point $t$ as an input to the Gated Recurrent Units for predicting the HF at the time point $t + 1$.

# 2 Related works

Until recently, HF prediction is made based on heuristic rules and a very limited number of biomarkers. Several types of HF prediction studies focused on readmission and survival for CHD patients. For example, LACE (Length of stay, Acuity of the admission, Co-morbidities, number of Emergency visits) index is used to predict patient readmission, and Meta-analysis Global Group in Chronic Heart Failure score (MAGGIC) is used to predict mortality. However, such evaluation metrics along with the widely used biomarkers such as natriuretic peptides often performed poorly when applied to different cohorts [9–11].

RNNs have emerged as a powerful machine learning approach to model longitudinal medical data. RNNs with long-short-term-memory (LSTM) unit or gated recurrent unit (GRU) are especially effective in capturing long range non-linear dependencies in a sequential event [12, 13]. For example, [6] applied LSTM to pediatric patients from Intensive Care Unit (ICU) to predict the diagnostic code of multiple diseases; [5] developed Doctor AI that used GRU to model the diagnoses and medication codes from EHR data to predict the diagnoses at the patients' next visit; [14] combined latent topic mixture of each patient inferred from their diagnostic code with RNN to predict patients' ICU readmission; [15] used similar topic+RNN for sequential diagnoses prediction but infers topic mixture from multi-modal EHR data.

A few deep learning models were also applied to HF predictions. For the short-term hospital re-admission, [4] applied multi-layer perceptron to predict readmission of HF patients in 30 days; [16] applied several deep neural networks to predict 30-day readmission for the CHD patients; [17] used a LSTM model to predict HF based on patients' recent history within 1.5 year. However, to the best of our knowledge, very few method attempted to model *the long-term progression of CHD* for large patient cohort over 20 years of medical follow-up. Additionally, very few method is able to perform multi-task prediction of both the HF events and other CHD-related co-morbidity variables, which also change during patients' life-time.

## 3 Data

### 3.1 Quebec congenital heart disease database

The dataset is derived from the EHR documented Quebec Congenital Heart Disease database with 84498 patients and 28 years of follow-up from 1983 to 2010 [18, 19]. The dataset is composed of data from 3 sources: demographics and vital status, inpatient and outpatient diagnoses, and surgery history. For each patient, the patient history can be traced back to the later of birth or 1983, and followed up to the end of the study or death of the patient, whichever came earlier.

We define HF events as the hospitalizations with HF being the admission and/or discharge diagnoses. For the HF study, only patients with at least one HF event were selected. Within the Quebec CHD database, a total of 9160 patients have had an HF history. Among them, nearly half (47.13%) had one heart failure hospitalization (HFH). One fifth had two HFHs and one tenth had three HFHs. About 15% of the patients had 5 or more HFHs (S1 Fig). Regarding age at the first HFH, the mean and median values were 61.80 and 68.26, respectively, with an inter-quartile of 54.82—77.13.

All patients were assigned 1 or 2 CHD diagnoses using a previously described and validated hierarchical algorithm [18]. Severity of CHD was classified on the basis of anatomic diagnosis into 4 types: severe, shunts, valvular, and unspecified CHD lesions (S1 Table). Severe lesions were with the highest probability of being associated with cyanosis at birth among the 4 lesion types. Here we used one-hot-encoding to encode the lesion type as a four binary variable.

In addition, we also used 11 CM diagnoses, which were recorded for each patient since their onsets. Among these 11 CM diagnoses, there are 3 CMs that areacute, namely "Acute myocardial infarction", "Infective endocarditis", and "Sepsis". We treated the other 8 CM variables as static in our model because we do not have their measurement at every time point for each patient.

We also included age, sex (male and female) and histories of surgery in our analysis and for training the RNN model. The original surgical operations belong to one of the four complexity categories [20, 21] (S3 Table). Due to the very low frequency of surgeries records across all the four complex levels, we combined the four types of surgeries into a single variable to overcome the sparsity and to have a meaningful estimation of its effect on HF.

In summary, we utilized 20 variables in our model: age (continuous variable), 2 variables corresponding to sex at birth, 4 one-hot-encoded variables for the lesion types, 11 binary CM variables, a single surgery variable corresponding to any of the four possible surgery complex levels, and a variable indicating HF at the previous time step.

### 3.2 Statistical analysis

We compared patients with and without HF in terms of the proportion of death, lesions, and the 11 CMs. As expected, the proportion of deceased patients is higher among the patients with HF than the patients without HF. Among the four types of CHD lesions, shunt is much

**Table 1. Comparison in selected demographic and clinical characteristics between CHD patients with and without HF.**

| Co-Morbidity | HF+ N = 9160 | HF- N = 74953 | Log Ratio | p-value |
|---|---|---|---|---|
| Death | 3557 (38.8%) | 5066 (6.8%) | 2.52 | <0.001 |
| CHD Lesion | | | | |
| Severe lesion | 970 (10.6%) | 8530 (11.4%) | -0.1 | 0.0171 |
| Shunt lesion | 2025 (22.1%) | 40445 (54.0%) | -1.29 | <0.001 |
| Valve lesion | 1811 (19.8%) | 11754 (15.7%) | 0.33 | <0.001 |
| Other lesion | 4354 (47.5%) | 14224 (19.0%) | 1.32 | <0.001 |
| Acute myocardial infarction | 2724 (29.7%) | 2714 (3.6%) | 3.04 | <0.001 |
| Coronary artery disease | 5447 (59.5%) | 6986 (9.3%) | 2.67 | <0.001 |
| Arrhythmia | 4155 (45.4%) | 3699 (4.9%) | 3.2 | <0.001 |
| Ventricular arrhythmias | 992 (10.8%) | 846 (1.1%) | 3.26 | <0.001 |
| Pulmonary hypertension | 2771 (30.3%) | 2182 (2.9%) | 3.38 | <0.001 |
| Infective endocarditis | 637 (7.0%) | 837 (1.1%) | 2.64 | <0.001 |
| Diabetes | 2799 (30.6%) | 3490 (4.7%) | 2.71 | <0.001 |
| Stroke | 2215 (24.2%) | 3610 (4.8%) | 2.33 | <0.001 |
| Chronic liver disease | 557 (6.1%) | 765 (1.0%) | 2.57 | <0.001 |
| Chronic kidney disease | 2788 (30.4%) | 1717 (2.3%) | 3.73 | <0.001 |
| Sepsis | 988 (10.8%) | 1923 (2.6%) | 2.07 | <0.001 |

The p-values were generated by binomial testwith background rate set to 9160/(9160+74953) = 0.109.

less frequent among patients with HF than those without HF. We also observe a significant enrichment of the 11 CM variables for patients with HF (Table 1), indicating the informativeness of these variables in predicting HF.

### 3.3 Selection of age group and positive cases

Because most of the HFHs take place after age 40 and the disease profile is quite different between adults and children, we chose to model only the HFHs after age of 40 for each patient. This yielded 8093patients with an average 71.4 time points(equivalent to 35.7 years).

### 3.4 Synthetic data

In order to ensure anonymity and comply with the data access conditions, we are unable to publicly release the records used to train the model. Instead, we createda synthetic dataset forusers to test-run our code. Specifically, we sampled each of the 11 CM variables along with HFH based on their observed frequency in the realdataset such that the simulated data preserves approximately the same distribution as the original data. These data are available at https://github.com/li-lab-mcgill/recurrent-disease-progression-networks.

## 4 Methods

### 4.1 Gated recurrent units architecture

The basic unit in our model is the gated recurrent unit (GRU), which is considered as an ablated version of LSTM as it eliminates the output gate and is twice faster than LSTM [22]. The update gate of the GRU (a sigmoid function $z_t = \sigma(W_z[h_{t-1}, x_t])$) takes previous state $h_{t-1}$ and current input $x_t$ and controls how much past information to keep. Justlike the LSTM, the update gate deals with the vanishing gradient problem for long sequential events. This is crucialin order to model up to 28 years of patient history in our data.

The reset gate defined as $r_t = \sigma(W_z[m_{t-1}, x_t])$ controls how much information from the past to underline{discard}. The current memory defined as $\tilde{h}_t = \tanh(W_h[r_t h_{t-1}, x_t])$ is ahyperbolic tangent function that depends on the reset gate. The final memory as a linear combination of past state and current memory is the state passed to the next step. It is defined as $h_t = (1 - z_t)h_{t-1} + z_t\tilde{h}_t$. If the output from the update gate $z_t$ is small, then the unit tends to use the past information, and vice versa.

## 4.2 Deep Heart-Failure Trajectory Models (DHTMs)

**4.2.1 Training the DHTM and DHTM+C models.** We developed an RNN model called Deep heart-failure trajectory (DHTM) to predict HF at the next time step $t \geq 2$ based on the information at the earlier time points $1 \leq i \leq t$ (Fig 1). Since the RNN model takes the discrete time series, we divided the patient records into time steps with six-month interval. At each time step, the RNN input is a 20-dimension integer vector, composed of age as integer, 2 one-hot-encoded sex variables, 4 one-hot-encoded lesion types, all 11 CM variables, and surgery count.

To not clutter the notation, we consider a single patient first. At time point $i \in \{1, \ldots, T\}$ for T time points, we denote $y_i$ and $\mathbf{c}_i$ as the binary HF event and the vector of input features, respectively. The input feature $\mathbf{c}_i$ and HF $y_i$ are first passed to a dense layer, then a number of consecutive GRU layers, and finally a sigmoid layer for the output $\hat{y}_{i+1}$ as the HF prediction at time $i + 1$. The number of GRU layers was chosen based on empirical evaluation on a validation set. To effectively train the DHTM, we used recently developed neural network techniques namely residual connections [23] and layer normalization [24].

Since knowing the CM variables at time point $i$ helps in predicting HF at $i + 1$, we propose a second model called DHTM+C (Fig 2). Here we predict *both the HF and CM at time point $i$* and use the predicted HF and CM as input to the recurrent unit in order to predict HF at $i + 1$. The input CM $\mathbf{c}_i$ and HF $y_i$ are passed through the first dense (i.e., fully connected), GRU, and a sigmoid layer to generate predicted CM $\hat{\mathbf{c}}_{i+1}$ at time $i + 1$. The same output is then passed to the second dense and GRU layers. The output from the two GRU layers and the output of the first sigmoid layer are then concatenated and passed to the second sigmoid layer, and the result is the predicted HF $\hat{y}_{i+1}$. Among the 11 CM diagnoses, there are 3 CMs that areacute, namely

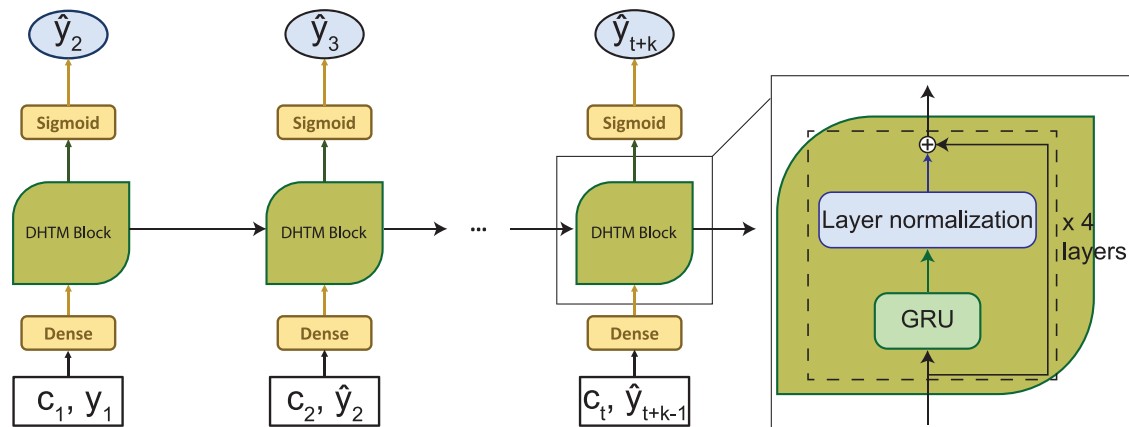

**Fig 1. Deep Heart-failure Trajectory (DHTM) Model.** During the training, DHTM learns to predict the next time point using observed acute CMs $c_i$ and HF $y_i$ for each patient. After training, DHTM can make a trajectory prediction using the acute CMs at the last observed time point $c_t$ and its own predicted HFH $\hat{y}_{t+i}$ as input for predicting HFH $\hat{y}_{t+i+1}$ at time point $t + i + 1$, where $i$ is the number of predicted time points so far. Thisauto-regressive process continues in an arbitrary number of future time points.

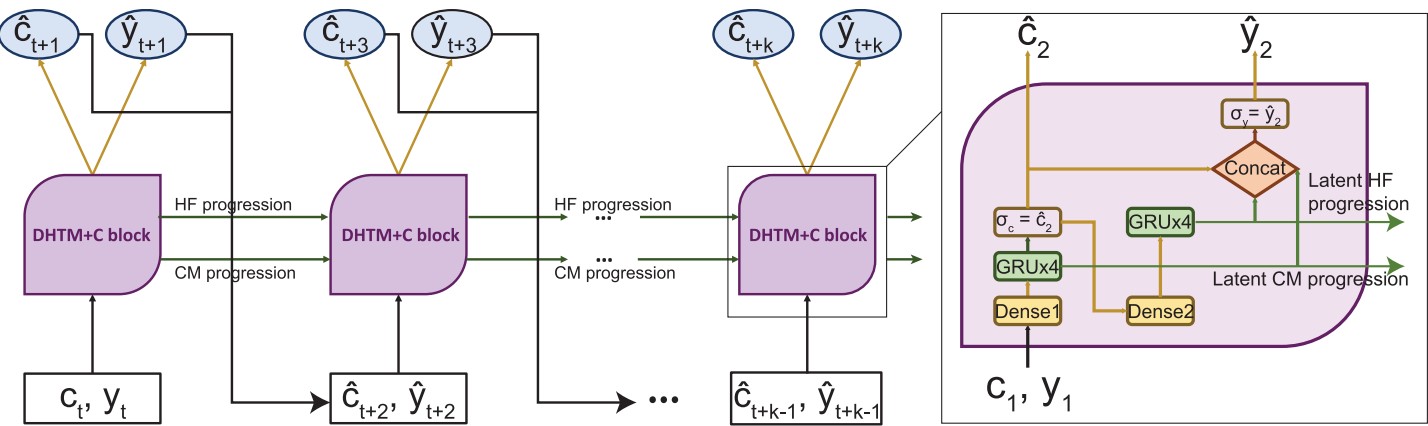

**Fig 2. Deep heart-failure trajectory plus co-morbidity (DHTM+C) model.** The overall designed of DHTM+C is similar to the DHTM illustrated in Fig 1 except that DHTM+C learns to predict both HFH and CM jointly during the training. During the trajectory prediction, the DHTM+C model takes the last observed CM and predicts both the HFH and CM at time point t+1. It then takes these predicted values as input at time point $t + 2$ to predict the HFH and CM at time point $t + 3$.

"Acute myocardial infarction", "Infective endocarditis", and "Sepsis". Therefore, we predict these 3 acute CMs while fixing the other 8 CMs throughout the model training. In particular, we add the prediction loss of the CMs at every time step as an auxiliary loss term to the prediction loss of HF. This ensures that the model is updated more frequently by the gradients of the total loss (sum of the HF loss and the CM loss) at every time step. To verify that, we observed that the CM loss decreased during each training epoch (S7 Fig). As a result, DHTM+C is able to model the progression of HF risk and learn a stable representation of the changes in the 3 acute CMs at every time step.

**4.2.2 HF trajectory prediction.** To predict HF at the $(t + 1)^{th}$ time point, our DHTM model takes as input the HF events and the input features observed up to and including $t^{th}$ time points. It builds up the hidden trajectory over the $t$ time points. It then fixes the input features observed at the last $t^{th}$ time point to make subsequent trajectory prediction of HF (Fig 1). To predict HF at time $t + i + 1$, the input to DHTM at time $t + i$ is the predicted HF at time $t + i$, $\hat{y}_{t+i}$. In contrast, because DHTM+C is able to predict both the 3 acute CM variables and HF at each time point t+i, it uses them as input to predict the next CM $\hat{c}_{t+i+1}$ and the next HF $\hat{y}_{t+i+1}$ (Fig 2). In principle, both DHTM and DHTM+C allow us to contiguously make risk prediction in an infinite horizon based only on a finite number of $t$ time points for a given patient.

**4.2.3 Objective function.** Since the model strictly outputs a probability of HF through a sigmoid activation $\sigma(x) = \frac{1}{1+e^{-x}}$, we minimize the cross-entropy as the loss function. Given a ground truth label $y_{p,i}$ and the model prediction $\hat{y}_{p,i}$ for patient $p$ and time $i$, we have the total loss as:

$$C(y, \hat{y}) = \sum_{p,i} - y_{p,i} \log(\hat{y}_{p,i}) - (1 - y_{p,i}) \log(1 - \hat{y}_{p,i}) \tag{1}$$

However, the model penalizes equally positive and negative labels. Since each patient will only have a few recorded HF hospitalizations throughout their entire life, the dataset label is heavily skewed towards negative examples, which may mislead the model to achieve a high accuracy at the cost ofprecision. Thus, we use the $\alpha$-weighted focal loss function, which was

previously used in object detection for predicting class-imbalanced labels [25]:

$$F(y \mid \hat{y}, \alpha, \beta) = \sum_{p,i} -\alpha(1 - \hat{y}_{p,i})^{\beta} y_{p,i} \log(\hat{y}_{p,i}) - (1 - \alpha)\hat{y}_{p,i}^{\beta}(1 - y_{p,i}) \log(1 - \hat{y}_{p,i}) \qquad (2)$$

where $\alpha$ and $\beta$ were respectively set to 0.25 and 2 based on [25].

We represent both HF and the acute CMs as a single multi-output binary cross-entropy objective, where $\mathbf{y}_{p,i}$ is a multi-label vector and $\hat{\mathbf{y}}_{p,i}$ is a sigmoid-activated vector representing the predicted progression in HF risk and the acute CMs. Thus, Eqs 1 and 2 are applicable to both DHTM (where the $y$ and $\hat{y}$ will be a scalar for a given time step) and DHTM+C (where $y$ and $\hat{y}$ are vectors).

**4.2.4 Mitigating vanishing gradients across layers.** By design, GRUs are capable of mitigating vanishing gradients caused by backpropagation *through all of the time points* [22]. However, in order to build deeper GRU models, we also needed to ensure that our models can correctly backpropagate gradients *through all of the layers*. For the DHTM model, we introduced residual connections and layer normalization to mitigate vanishing gradients across layers. Residual connections was implemented in the transformer architecture in deep models with up to 12 hidden layers [26]. Layer normalization mitigates the effect of exploding/vanishing gradients caused by the magnitude of the recurrent units. For the DHTM+C model, because of the more frequent feedback produced by the prediction loss of the 3 CM and HF, we did not implement the residual connections and layer normalization.

## 4.3 Baseline methods

To compare the performance of our DHTMs, we trained four baseline models, namely Logistic Regression, Support Vector Machine (SVM) with a linear kernel, LSTM and Cox regression. Both logistic regression and SVM are trained to predict HF at time step $t$ based on the inputs at the last recorded time step ($X_{t-1}, y_{t-1}$). We also evaluated the standard LSTM model, which does not use the residual connections [23] and layer normalization [24] techniques as in our DHTM model nor the modified GRU architecture as in our DHTM+C model, but uses the same objective function as in Eq (2).

For the Cox regression, we performed standard survival analysis using the Cox-based Andersen-Gill (AG) model for recurrent event analysis [27]. The AG model analyzed the recurrent HF data in continuous time. Each patient has multiple observations—each starting from the previous event (or the start of follow-up in the case of first event) to the next event (or the end of follow-up in the case of last event). The AG model was adjusted for covariates in the same time-dependent manner as in the RNN model. In addition, we included previous number of HF hospitalizations as a covariate. Robust variance estimator was used to account for the correlation between observations from the same patient. Using the AG model, we were able to generate estimated hazard ratios (HR) for covariates and predicted survival probabilities for each patient within any time interval. However, the prediction may not be directly comparable to other models due to the different structure of the training data.

## 4.4 Choices of threshold for HF trajectory prediction

Each method has different distribution over the positive and negative HF events (S6 Fig). Therefore, choosing the thresholds that has good generalizability is essential for the HF models to be useful in practice. To this end, we tested 4 thresholds. Let $p_1, p_2, \ldots, p_k$ be the scores computed by a model for the $k$ time points for all of the HF events across all of the time points over all of the patients in the training set. Let $n_1, \ldots, n_m$ be the scores for the $m$ time points across all of the non-HF events. In total, there are $m + k$ time points in the training dataset. Then, we

computed the means of HF scores and non-HF scores as $\bar{p} = (p_1 + p_2 + \cdots + p_k)/k$ and $\bar{n} = (n_1 + n_2 + \cdots + n_m)/m$, respectively. The four thresholds are defined below and evaluated for each method in terms of F1 Score (S5 Fig). Unless mentioned otherwise, the thresholds were chosen based on the full training data (i.e., 75% of the total patients).

1. **Frequency threshold** is calculated as the total number of positive HF-events divided by the total number of time points over all patients in the training set: $k/(m + k)$;

2. **Conservative threshold** is calculated as the average risk score of the true HF events in the training set: $\bar{p}$. This threshold is the most relevant in a resource constrained context and clinically meaningful as the model identifies the high risk patients that are at need for immediate medical interventions;

3. **Balanced threshold**. To increase the sensitivity in non-resource constrained situations, we calculate the midpoint between the average risk score of the true HF events, and the average risk score of the non-HF time steps: $(\bar{p} + \bar{n})/2$;

4. **Optimized threshold**. This is a commonly used threshold in the machine learning community. The threshold is found via a grid search that maximizes the F1 Score based on the validation set. We trained each model on the training set, which is 63.75% of the total patients. We set aside 11.25% of the total patients as the validation set to select the best threshold that gives the highest validation F1 score for each method. Therefore, we used in total 75% of the data to train and choose thresholds for each deep recurrent model. We then evaluated each model based on their optimized threshold on the test set, which is 25% of the total patients.

## 4.5 Evaluation

We evaluated two prediction tasks: (1) next time point prediction; (2) trajectory prediction. For task 1, we used the observed $t$ time points to predict the $t + 1$ time points for each patient $p$, where $t \in \{1, \ldots, T_p - 1\}$, for all of the time points $T_p - 1$ time points. For task 2, each RNN model was given 15 years of records on CM changes and HFH, starting from the age of 40.

In both tasks, we trained all models on 75% of the patients, and used the remaining 25% for evaluation. The baseline logistic regression was trained with L2-norm regularization, and the SVM was trained using a penalization parameter of $C = 1$. The penalties of both logistic regression and SVM were selected based on a thorough grid search over the regularization parameter.

For task 1, we evaluated all the models using the area under the receiver-operating characteristic curve (AUROC), and the area under the precision-recall curve (AUPRC) metric. Both metric are based on the unrolled patient records. For task 2, we used AUROC, AUPRC, and F1 Score ($\frac{2 \times \text{Precision} \times \text{Recall}}{\text{Precision} + \text{Recall}}$) based on systematically determined thresholds (Section 4.4).

## 5 Results

### 5.1 DHTM outperforms baseline models in predicting next HF

We sought to evaluate our proposed models' ability to predict HF at the next time point using all of the earlier time points. To this end, we generated the ROC and precision-recall (PR) curves for our proposed DHTM and DHTM+C and the 3 baseline models, and we evaluated the accuracy in terms of AUROC and AUPRC (S3 Fig). In both metric, DHTM and DHTM +C achieved the highest AUROC of 0.8626 and 0.8595, respectively whereas standard LSTM obtained 0.8582 AUROC—slightly worse than our proposed DHTM.

DHTM+C and DHTM achieved AUPRC score of 0.2509 and 0.2446, whereas the standard LSTM is once again slightly worse than the DHTM models. At the 1% recall rate (i.e., number of predicted true positives over all of the positive HF events), our DHTM+C model achieved over 75% precision ranking the best model, and the DHTM model is the second best with 70% precision. In contrast, the static models namely SVM and logistic regressions achieve only 40% and 45% precision rate at the same recall rate, respectively. Interestingly, the Cox regression achieved the lowest AUPRC of 0.1096. It is possible that Cox regression is not suitable to the prediction task as it is designed for survival analysis and hypothesis testing over the covariates (i.e., co-morbidities) rather than risk prediction.

## 5.2 Trajectory predictions

We experimented four different thresholds in terms of the F1 scores for predicting HF trajectory (**Section 4.4**; S2 Table). Overall, the conservative and optimized thresholds produce higher F1 scores compared to the frequency and balanced thresholds (S5 Fig). Both the DHTM and DHTM+C consistently achieve higher F1 score compared to the baseline LSTM (Fig 3). With a conservative threshold, both our DHTM models are able to recall 50% of the positive HF events at the first year trajectory and nearly 20% of the positive HF at year 10 (S2b Fig). In contrast, LSTM can only recall 5% at year 1 and 1% at year 10. We also evaluated AUROC and AUPRC (S4 Table). DHTM+C demonstrates clear advantage over DHTM and LSTM models. As expected, both AUROC and AUPRC decrease for predicting HF in more distant future time points for each model because there are potentially many unobserved factors that may affect the patient outcome.

To gain further intuitionabout the trajectories predicted by each model, we plotted a randomly selected set of 5 patient exampleswith positive HF events (Fig 4). We observed that both the DHTM and DHTM+C predicted higher risk for these patients even at their earlier years before the HF onset compared to the standard LSTM model. Both DHTM and DHTM +C generated much more dynamic trajectories for these patients than the trajectories generated by LSTM. This is due to our better engineered RNN architecture that learns more

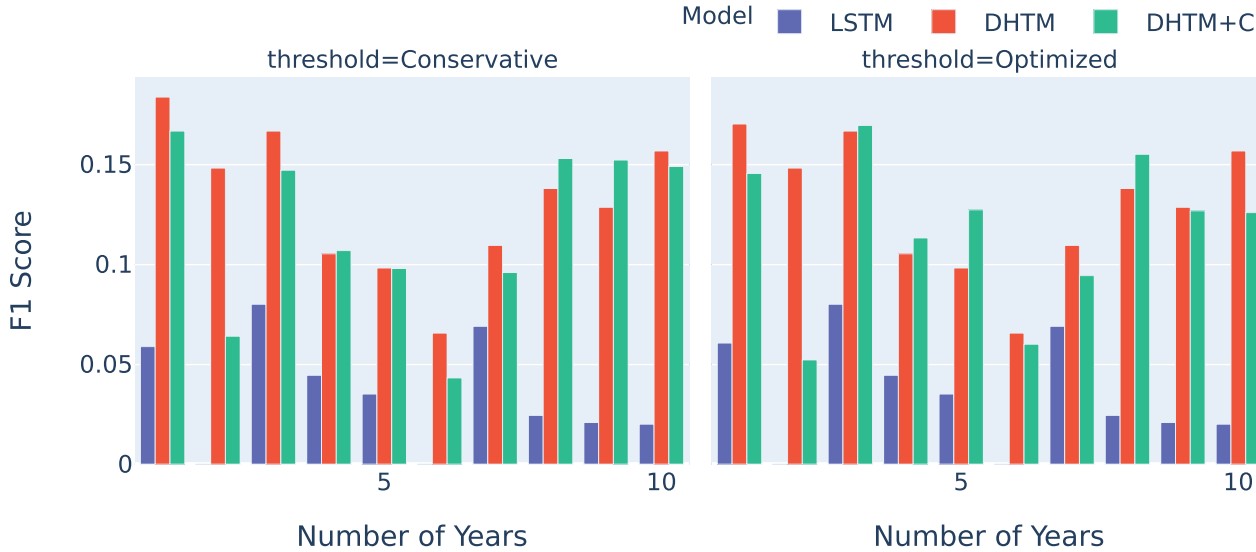

**Fig 3. F1 scores for the three recurrent architectures.** Each model was given 15 years of records on CM changes and HFH, starting from the age of 40. The F1 score as a function of years was then evaluated for each of the 3 RNN models on the test patients. The conservative (a) and optimized (b) methods were used to compute the thresholds. For evaluation of all of four thresholding approaches, please see S5 Fig.

meaningful trajectory information from other high-risk patients in making prediction of the new test patients.

### 5.3 HF-predictive input features learned by the RNN

Deep neural networks are often considered as "black box" machine learning models. Because of their distributed non-linear representation, interpreting the contribution of specific input (i.e., 20 input variables) is a challenge. We attempted to do so by examining the first dense layer connecting each of the 20 input units to each of the 64 hidden units using Hinton plot [28]. The rationale is that these weights control the very beginning of the hidden activities. The larger the weights the stronger impact the input units have on these hidden activities and vice versa. Interestingly, threeacute CM variables exhibit relative large weights for our DHTM+C and LSTM model which is consistent with medical observation (S4 Fig). However, the LSTM model *does not* predict these 3 CMs and the fact that these CMs still exhibit largest weights in LSTM is intriguing. On the other hand, our DHTM model appears to have a more distributed representation over all of the20 input variables, implying a distinct learning behaviour compared to the other two networks. More advanced techniques such as integrated gradients [29] may reveal more interesting patterns, which we will leave as future work.

## 6 Discussion

One of the goals ofprecision medicine is to develop intelligent tools that help prognoses based on the phenotypic characteristics and unique medical history of individual patients. In this paper, we present a deep learning approach called Deep Heart Trajectory Model (DHTM) as a step towards theprecision prognosis for CHD patients. Among the facets of machine learning models, recurrent neural network stands out as natural choice in this application because of its capacity to model long sequential events. With the recent advancement in RNN architectures such as LSTM and GRU, RNNs often perform competitively compared to other approaches in terms of capturing long-range contextual information. This is very important in our application of HF trajectory prediction, where we are modeling up to 28-year of medical history of each patient. It is worth noting that our contribution in this paper goes beyond the application of GRU and LSTM, We made several novel design to the network architecture combined with advance deep learning techniques such as residual connection and layer normalization to achieve the state-of-the-art performance. In particular, our model jointly predict HF as well as multiple CM to achieve more accurate prediction on the future trajectory of HF.

We have shown several examples of how our model predicts HF trajectories in different scenarios (Fig 4). For patients with high HF risk profile, both DHTM and DHTM+C successfully predict their risk trajectories. Nonetheless, there are still room for further improvements. With a longer time frame for prediction, some of the patients' CM may change more dramatically. By the nature of the CM studied, many of them are either lifelong (since birth) or chronic after diagnosed (since disease onset). The change in CM occurs only when a patient has been diagnosed with certain CM or a surgery has been performed. Thus, the model is inclined to predict no change for these CM. This is consistent with our data, which contain only disease diagnoses but no clinical data for marking the disease progression of the CM especially for lifelong and chronic conditions. We have chosen the most powerful CMs among candidate predictors based on literature and our previous publications on HF [30–32]. The majority of the CMs are lifelong conditions since birth such as congenital heart defects or chronic conditions since the first onset such as diabetes. For these CMs, there is no change in the data since birth or the first diagnosis. As for the acute CMs that might have new onsets, the rate of recurrent events is very low. For example, only 3.03% of the patients have recurrent sepsis and 1.67% have

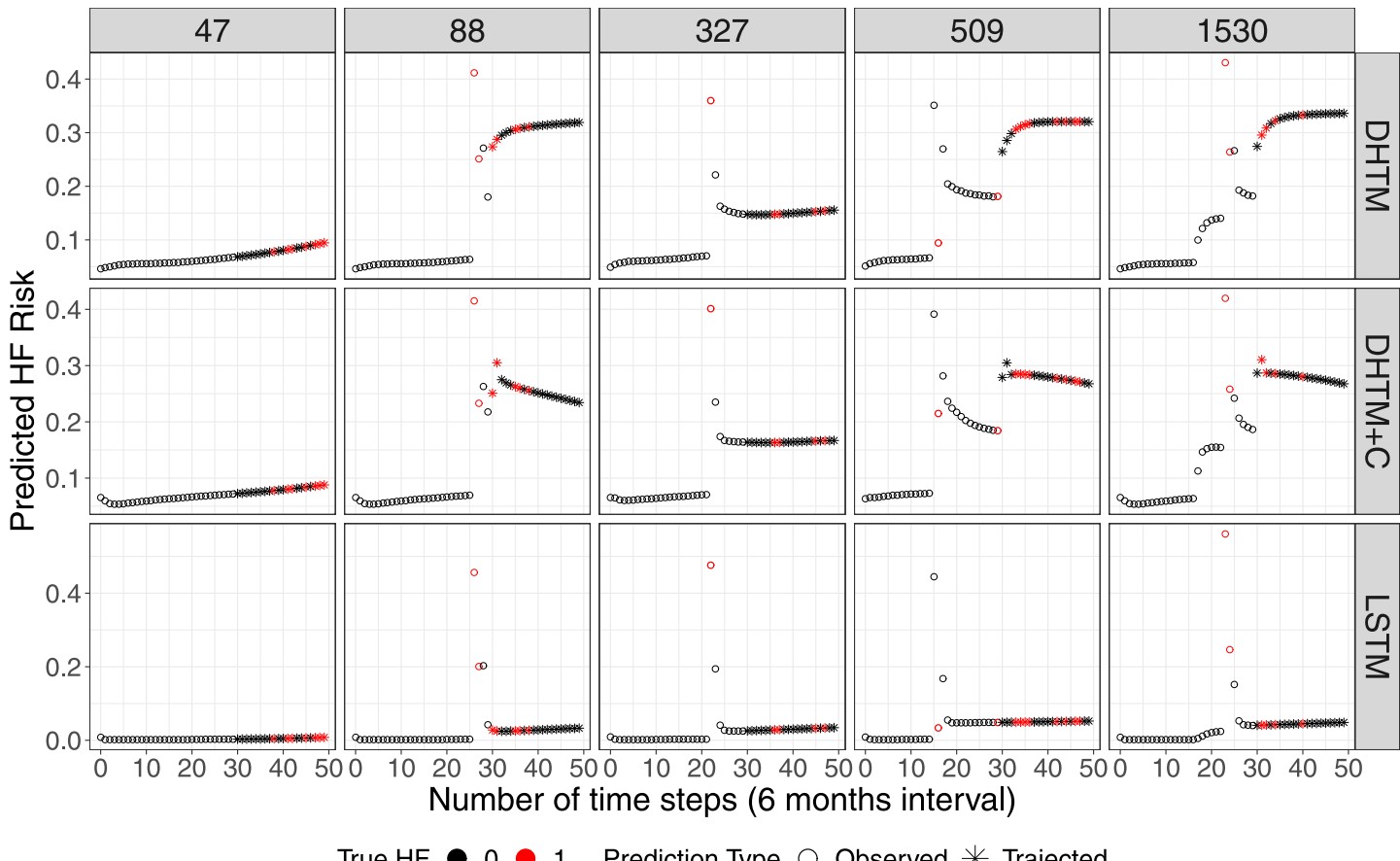

**Fig 4. Five patient trajectory examples predicted by the 3 RNN models.** Each panel illustrates the predicted HF risk at the observed time points (circles) and at the trajected time points (asterisks). Black and red color indicates true negative and positive HF event, respectively. The difference in performance between DHTM/DHTM +C and LSTM is consistent with Fig 3.

recurrent infectious endocarditis. On the other hand, this is also due to the limitation of the data source being administrative data. We are currently in the process of requesting clinical data for CHD patients and will include them in the next model of HF trajectories. Due to the same limitation, the model gives a less accurate prediction for a longer prediction trajectory. A general trend we observed is that the model may give a more drastic change in predicted probability during the early part of the prediction, but not for the later part of the prediction. The model could be improved by having a more accurate CM prediction, adding clinical data on the progression of CM, and supplemented with more relevant CM.

## 7 Conclusion

To our knowledge, this is the first study to model long-term and short-term HF trajectories in patients with CHD. We present an RNN model for HF trajectory prediction based on patient's HF and CM records. We have demonstrated that the proposed model outperforms various baseline models and can predict different types of trajectories. Our future direction includes further investigation of the model design of the RNN architecture such as weighting the hidden activities by the time gap, given that clinical data from outpatient visits are irregularly sampled. We are also interested in comparing the result with the predictions from physicians and polishing the model based on their feedback. More features may be added to the feature set,

such as the International Classification of Diseases (ICD) codes which provides a systematic classification of diagnoses for various diseases and clinical data such as lab test results and imaging data. We believe that expanding the feature set could improve the performance. It should be noted that CHD patients have different disease mechanisms and treatment for HF from the general population. Thus, the clinical interpretation of the prediction models for HF constructed among CHD patients could not be generalized to the general population. However, the model we demonstrated in this paper can be readily adapted to predict other chronic or acute conditions. In conclusion, we believe that our model is a significant step towards providing aid to physicians with an accurate prediction of HF trajectory and ultimately informing allocation of health services to prevent HF in patients at high risk.

## Supporting information

**S1 Fig. Heart failure frequency distribution in our CHD dataset.**
(PDF)

**S2 Fig. Precision and recall scores for the conservative threshold.** Each RNN model was given 15 years of co-morbidities changes and heart failures, starting from the age of 40. The precision (a) and recall (b) scores as a function of years were then evaluated for each of the 3 RNN on the test patients.
(PDF)

**S3 Fig. ROC and precision-recall in predicting next time heart failure onset.** We compared 5 different methods namely logistic regression, SVM, and three recurrent neural network approaches, which are RNN with LSTM recurrent unit, Deep Heart Trajectory Model (DHTM), and Single-time-step Model (STM). The overall precision-recall curve and the "zoom-in" view on the recall at 0.2 are displayed.
(PDF)

**S4 Fig. Hinton plots on each RNN model.** Hinton plots for the 3 RNN model weights in the first input-hidden layer. We plot from top to bottom the weights of the dense layer for DHTM +C, DHTM, and LSTM, respectively. For each panel, The rows are the 20 co-morbidity variables and the columns are the 64 hidden units. white and black color indicate positive and negative weights, respectively. The size of the squares is proportion to the magnitude of the connection weights between the input units and the hidden units.
(PDF)

**S5 Fig. F1 scores for our three recurrent architectures.** Each model was given 15 yearsof co-morbidities changes and heart failures, starting from the age of 40. The F1 scoreas a function of years was then evaluated for each of the 3 RNN on the test patients.
(PDF)

**S6 Fig. Distributions of risk scores predicted by each of our RNN models.** Whereas LSTM collapses at 0, both the DHTM+C and DHTM models spread the risk score over a greater range. For this reason, a different threshold is used for every model, as displayed in S2 Table.
(PDF)

**S7 Fig. Loss history of the co-morbities prediction task in the DHTM+C model.** The model was trained for 30 epochs, but quickly decreased to 0.0005 due to the sparsity of the labels.
(PDF)

**S1 Table. Patient distribution in congenital heart defects.**
(PDF)

**S2 Table. The value of the thresholds for each of the four methods.** A threshold is applied for every patients and every time step for a given model.
(PDF)

**S3 Table. Complex levels of cardiac surgical operations in patients with congenital heart disease.**
(PDF)

**S4 Table. Table of area under the receiver operating characteristic curve (AUROC) area under the precision-recall curve (AUPRC) for every year in the trajectory prediction.** Each RNN model was given 15 years of co-morbidities changes and heart failures, starting from the age of 40. They then predicted the next 10 years of HF events.
(PDF)

## Author Contributions

**Conceptualization:** Xing Han Lu, Aihua Liu, Shih-Chieh Fuh, Liming Guo, Yi Yang, Ariane Marelli, Yue Li.

**Data curation:** Xing Han Lu, Liming Guo.

**Formal analysis:** Xing Han Lu, Yi Lian.

**Investigation:** Xing Han Lu, Aihua Liu, Yue Li.

**Methodology:** Xing Han Lu, Aihua Liu, Shih-Chieh Fuh, Yi Lian, Liming Guo, Ariane Marelli, Yue Li.

**Project administration:** Xing Han Lu, Yi Yang, Ariane Marelli, Yue Li.

**Resources:** Xing Han Lu, Aihua Liu, Liming Guo, Ariane Marelli, Yue Li.

**Software:** Xing Han Lu.

**Supervision:** Yi Yang, Ariane Marelli, Yue Li.

**Validation:** Xing Han Lu.

**Visualization:** Xing Han Lu, Shih-Chieh Fuh.

**Writing – original draft:** Xing Han Lu, Aihua Liu, Shih-Chieh Fuh, Yi Lian, Yue Li.

**Writing – review & editing:** Xing Han Lu, Aihua Liu, Yue Li.

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
