## [Decision Letter · Decision Letter 0]

16 Oct 2020

PONE-D-20-20231

Recurrent disease progression networks for modelling risk trajectory of heart failure

PLOS ONE

Dear Dr. Li,

Thank you for submitting your manuscript to PLOS ONE. After careful consideration, we feel that it has merit but does not fully meet PLOS ONE’s publication criteria as it currently stands. Therefore, we invite you to submit a revised version of the manuscript that addresses the points raised during the review process.

Please note that both reviewer found your manuscript and your method interesting but raised several points to be addressed. The applies particularly to reviewer #1. In addition to the reviewers comments, I would like you to consider reporting the age at diagnosis. You should also have your article read by a clinician that use the correct medical terminology (see comments reviewer #2). Furthermore, it is not entirely clear to me if you only included patients with congenital heart diseases as cause of HF. It seems to be the case. Therefore, you should be clear also in the title that you refer to this kind of patients with HF. They may be very different from other patients with HF. You also need to define the lesions of CHD. I also miss a description of the number and time course of events in the patient group.

We look forward to receiving your revised manuscript.

Kind regards,

Hans-Peter Brunner-La Rocca, M.D.

Academic Editor

PLOS ONE

Journal Requirements:

2. We note in your Data Availability statement you have advised "No - some restrictions will apply" and also stated "All relevant data are within the manuscript and its Supporting Information files."

If there are ethical or legal restrictions to sharing your data publicly, please explain these restrictions in detail. Please see our guidelines for more information on what we consider unacceptable restrictions to publicly sharing data: http://journals.plos.org/plosone/s/data-availability#loc-unacceptable-data-access-restrictions. Note that it is not acceptable for the authors to be the sole named individuals responsible for ensuring data access. We will update your Data Availability statement to reflect the information you provide in your cover letter.

Reviewers' comments:

Reviewer's Responses to Questions

**Comments to the Author**

1. Is the manuscript technically sound, and do the data support the conclusions?

Reviewer #1: Yes

Reviewer #2: Yes

2. Has the statistical analysis been performed appropriately and rigorously? 

Reviewer #1: N/A

Reviewer #2: Yes

3. Have the authors made all data underlying the findings in their manuscript fully available?

Reviewer #1: No

Reviewer #2: Yes

4. Is the manuscript presented in an intelligible fashion and written in standard English?

Reviewer #1: Yes

Reviewer #2: Yes

5. Review Comments to the Author

Reviewer #1: This manuscript reports a study of predicting HF events using RNN models. The authors designed two RNNs, i.e., DHTM and DHTM+C, both of which are based on the GRU architecture. Then the authors trained the models on an EHR dataset and evaluated the prediction performances. These models were also compared to the baseline models including logistic regression, SVM and a standard LSTM model. The results are promising, and the manuscript is well organized.

However, there are a number of problems with the manuscript, and there are also some concerns with the study:

1) The statements on p. 5, lines 146-149 implicitly contained the following two points:

a) In the DHTM model, the residual connections and layer normalization were added in order to lessen the vanishing gradient problem.

b) Adding CMs as targets of prediction in addition to HF in the DHTM+C model can lessen the vanishing gradient problem of the DTHM model.

However, both of the two points are problematic:

For a), the authors already described on p. 4 line 112 that the vanishing gradient problem is dealt with by the update gate in GRU models. Since DHTM model is based on GRU, why is vanishing gradient still a problem for DHTM model, which led to the addition of residual connection and layer normalization?

For b), why adding more targets of prediction can lessen the vanishing gradient problem?

2) The description of the objective functions (section 4.2.3) is confusing. Specifically,

a) In Eq. (1), the subscripts of y and yhat are missing.

b) In Eq. (2), both the summation and subscripts are missing.

c) It seems that both Eqs. (1) and (2) are used only the prediction of HF events. Since the DHTM+C model predicts multiple outcomes - both HF and CM events, what is the objective function defined for DHTM+C?

3) What is the objective function for the LSTM model? Is it the same as those in section 4.2.3, or a different one? If not the same, then the difference between the predictions of the LSTM model and the predictions of the DHTM/DHTM+C models may be explained by the difference in the objective functions.

4) For evaluation, precision and recall are included in addition to other metrics. However, directly comparing precisions (or recalls) of different models is unfair, since one can be optimized at the cost of the other. The authors should compare F-scores instead.

5) Precision and recall depend on a choice of thresholds. Why is the threshold chosen as the average of probabilities?

6) Did the authors choose different thresholds for different models, or one threshold for all models?

7) For LSTM model, what are the AP score and the precision at 1% recall? These two are reported for DHTM and DHTM+C models in section 5.1.

8) In section 5.1, it is stated that "Cox regression achieved the lowest prediction accuracy at 10% AP." What does "prediction accuracy at 10% AP" mean? How did the other models perform on this metric?

9) S2 Fig is exactly the same as Fig 3. Based on the description of S2 Fig, it seems that S2 Fig points to the wrong figure. In addition, it seems that the AUROCs should be reported in S2 Fig, but the description of S2 Fig only mentions AP and AUPRC.

10) The authors evaluated AUPRC and AUROC for the trajectory prediction for each year over 10 years, but I do not see these AUCs reported in Results.

11) On p. 7, line 199, it is stated that, for Task 2, the authors evaluated AUPRC and AUROC, while on the same page, line 206, it is stated that, for both tasks, the authors evaluated AUROC and AP. Then what is exactly evaluated for Task 2, AUPRC or AP or both? This is confusing.

12) For DHTM+C, are the evaluations done for only the HF prediction, or for all the CM predictions as well?

Reviewer #2: The authors of this study have made two heart failure prediction models (DHTM and DHTM+C) with recurrent neural networks based on GRU architecture. The dataset is derived from the EHR documented Quebec Congenital Heart Disease where they chose to model the prediction for heart failure in case of congenital heart disease, because heart failure origination from CHD would be better to predict. They chose data of adults from the age of 40 in order to exclude the disease profile of children. They ended up with data of 8093 patients of which they used 75% to train the neural network, and 25% to validate it. They used 16 covariates in the DHTM+C model to not only predict the heart failure events, but also onset of comorbidities and their effect of heart failure events. They compared the result of their GRU based RNN with baseline models LSTM, Logistic Regression, Support Vector Machine and Cox regression. The results showed that their two models outperformed the baseline models. These results are promising. However, the model is inclined to predict for no change for comorbidities, which could improve with further development.

1. On line 19 you state “Compared to HF induced by shock or chemicals...”, what is meant by shock? Shock is mostly a result of heart failure, not a cause.

2. On line 46 and 51 the listings are hard to read when using a period (.). Maybe suggest using semicolon (;).

3. On line 64 you describe HF events and history. How is heart failure and heart failure event defined? (LVEF <35%? or <40% or <50%? HFpEF? HFrEF?) You state that you include patients with at least one HF event. What kind of events? Acute heart failure events with need of admission? Were there patients with heart failure (<35% LVEF) but without "events"? If so, why were only patients with at least one event included? Do you think data bias could result from this?

4. On line 72 you state “surgery of four different complex levels”. Could you give an example of “severe” and “other” to make this more clear?

5. Table 1: Suggestion: make P values more readable by writing < 0.01 or < 0.01 or 0.01 or 0.34 for example.

6. On line 314 you state that "the model is inclined to predict for no change for these CM". So was the model not successful in predicting new onset om CM? Are there results on this?

7. Description of S3 Fig.: I assume in the sentence “While and black color indicate positive and negative weights, respectively.”, the word “While” should be “White”.

6. PLOS authors have the option to publish the peer review history of their article (what does this mean?). If published, this will include your full peer review and any attached files.

Reviewer #1: No

Reviewer #2: No

---

## [Author Response · Author response to Decision Letter 0]

1 Dec 2020

Please see DHTM _PLoS1_Response.docx

---

## [Decision Letter · Decision Letter 1]

18 Dec 2020

PONE-D-20-20231R1

Recurrent disease progression networks for modelling risk trajectory of heart failure

PLOS ONE

Dear Dr. Li,

Thank you for submitting your manuscript to PLOS ONE. After careful consideration, we feel that it has merit but does not fully meet PLOS ONE’s publication criteria as it currently stands. Therefore, we invite you to submit a revised version of the manuscript that addresses the points raised during the review process.

One of the reviewers have some remaining issues that I would like you to address.

We look forward to receiving your revised manuscript.

Kind regards,

Hans-Peter Brunner-La Rocca, M.D.

Academic Editor

PLOS ONE

Reviewers' comments:

Reviewer's Responses to Questions

**Comments to the Author**

1. If the authors have adequately addressed your comments raised in a previous round of review and you feel that this manuscript is now acceptable for publication, you may indicate that here to bypass the “Comments to the Author” section, enter your conflict of interest statement in the “Confidential to Editor” section, and submit your "Accept" recommendation.

Reviewer #1: (No Response)

Reviewer #2: All comments have been addressed

2. Is the manuscript technically sound, and do the data support the conclusions?

Reviewer #1: Yes

Reviewer #2: Yes

3. Has the statistical analysis been performed appropriately and rigorously? 

Reviewer #1: Yes

Reviewer #2: Yes

4. Have the authors made all data underlying the findings in their manuscript fully available?

Reviewer #1: Yes

Reviewer #2: Yes

5. Is the manuscript presented in an intelligible fashion and written in standard English?

Reviewer #1: Yes

Reviewer #2: Yes

6. Review Comments to the Author

Reviewer #1: The authors' responses and revisions have answered most of my questions. However, there are still some confusions.

1) The authors added a new section, i.e., Section 4.4, to describe the various choices of thresholds, but the notations there are very confusing.

a) The "frequency threshold" was defined as m/(m+k) and was stated in words as the number of positive labels divided by the total number of patients. So, it appears that m is the number of positive labels, i.e., the HF events, and (m+k) is the total number of patients. But these are in contradiction to what were described in the beginning of the section: m is defined as the number of time points for the non-HF events, i.e., the negative labels, and (m+k) is the total number of time points over all the patients (in the training set), not just the total number of patients (in the training set).

b) The authors defined p_bar = (p_1, p_2, ... , p_k)/k. Is p_bar is vector or scalar? It looks to me that p_bar, as an average of probabilities, should be defined as (p_1+p_2+...+p_k)/k. The same question exists for n_bar = (n_1, n_2, ... , n_m)/m.

2) For consistency in style, I would suggest rewriting the sentence "Interestingly, the Cox regression achieved the lowest prediction accuracy at 10% AUPRC" (in Section 5.1, 2nd paragraph) as "Interestingly, the Cox regression achieved the lowest AUPRC of 0.1096."

Reviewer #2: I thank the authors for their responses. With the new adaptations, I find the article, especially in terms of medical readability, much improved. The clarification of "acute comorbidities" also contributes to this. I only have a few suggestions to improve typo's and a choice of words:

Abstract: space missing between "acute" and "co-morbidities" in the sentence "HF and a set of acuteco-morbidities".

Line 25: space missing between "acute" and "co-morbidities" in the sentence "HF and a set of acuteco-morbidities".

Line 322: You write "personalized medicine", which is basically a correct term. However, precision medicine is being used more and more often nowadays, because some say that medicine is always "personalized". However, I leave this to the preference of the author.

7. PLOS authors have the option to publish the peer review history of their article (what does this mean?). If published, this will include your full peer review and any attached files.

Reviewer #1: No

Reviewer #2: No

---

## [Author Response · Author response to Decision Letter 1]

22 Dec 2020

Please see DHTM_PLoS1_Response2.pdf

---

## [Editor Report · Decision Letter 2]

23 Dec 2020

Recurrent disease progression networks for modelling risk trajectory of heart failure

PONE-D-20-20231R2

Dear Dr. Li,

We’re pleased to inform you that your manuscript has been judged scientifically suitable for publication and will be formally accepted for publication once it meets all outstanding technical requirements.

Kind regards,

Hans-Peter Brunner-La Rocca, M.D.

Academic Editor

PLOS ONE
---

## [Editor Report · Acceptance letter]

28 Dec 2020

PONE-D-20-20231R2 

Recurrent disease progression networks for modelling risk trajectory of heart failure  

Dear Dr. Li:

I'm pleased to inform you that your manuscript has been deemed suitable for publication in PLOS ONE. Congratulations! Your manuscript is now with our production department. 

Kind regards, 

on behalf of

Dr. Hans-Peter Brunner-La Rocca 

Academic Editor

PLOS ONE